# Clinical Predictors of Gastrointestinal Bleeding Source before Computed Tomography Angiography

**DOI:** 10.3390/jcm12247696

**Published:** 2023-12-15

**Authors:** Wisam Sbeit, Maamoun Basheer, Amir Shahin, Sharbel Khoury, Botros Msheael, Nimer Assy, Tawfik Khoury

**Affiliations:** 1Department of Gastroenterology, Galilee Medical Center, Nahariya 221001, Israel; wisams@gmc.ac.il (W.S.); amirs3@gmc.gov.il (A.S.); tawfikkhoury1@hotmail.com (T.K.); 2Azrieli Faculty of Medicine, Bar-Ilan University, Safed 1311502, Israel; 3Department of Radiology, Galilee Medical Center, Nahariya 221001, Israel; sharbelk@gmc.gov.il (S.K.); botrosm@gmc.gov.il (B.M.); 4Internal Medicine Department, Galilee Medical Center, Nahariya 221001, Israel; nimera@gmc.gov.il

**Keywords:** CT, Angio, bleeding, gastrointestinal, predictors

## Abstract

Background: Acute gastrointestinal bleeding (GIB) is a commonly encountered medical emergency. In cases of negative endoscopic evaluations, computed tomography angiography (CTA) is usually the next diagnostic step. To date, data regarding positive CTA examinations are lacking. We aimed to assess the clinical and laboratory parameters that predict a positive CTA examination, as demonstrated by the extravasation of contrast material into the bowel lumen. Methods: We performed a single-center retrospective study, including all patients who were admitted with GIB and who underwent CTA. Analysis was performed to compare patients’ characteristics, and logistic regression was used to explore parameters associated with a positive CTA. Results: We included 154 patients. Of them, 25 patients (16.2%) had active GIB on CTA vs. 129 patients (83.8%) who did not. On univariate analysis, several parameters were positively associated with active GIB, including congestive heart failure (OR 2.47, 95% CI 1.04–5.86, *p* = 0.04), warfarin use (OR 4.76, 95% CI 1.49–15.21, *p* = 0.008), higher INR (OR 1.33, 1.04–1.69, *p* = 0.02), and low albumin level (OR 0.37, 95% CI 0.17–0.79, *p* = 0.01). On multivariate logistic regression analysis, only high INR (OR 1.34, 95% CI 1.02–1.76, *p* = 0.03) and low albumin (OR 0.3, 95% CI 0.12–0.7, *p* = 0.005) kept their positive association with active bleeding, while a high ASA score was negatively associated with an active GIB. Conclusions: We could identify high INR and low albumin as strong predictors of active GIB, as demonstrated by positive CTA. On the other hand, comorbid patients classified by a high ASA score did not experience a higher rate of active GIB.

## 1. Introduction

Acute gastrointestinal bleeding (GIB) constitutes a major cause of hospital admissions worldwide. It is estimated that the annual incidence of upper GIB is 100/100,000 [1,2,3], while the annual incidence of lower GIB is about 36/100,000 residents [4]. Actually, most bleeding episodes stop spontaneously, but in cases of ongoing bleeding, the mortality rate approaches 14% for acute upper GIB [3,5,6], and up to 3.9% for acute lower GIB cases [4]. In cases of upper GIB, both the American and European guidelines recommend early upper GI endoscopy (<24 h) following hemodynamic resuscitation and to consider very early (<12 h) upper GI endoscopy in patients with persistent hemodynamic instability (tachycardia, hypotension) despite ongoing attempts at volume resuscitation; in-hospital bloody emesis/nasogastric aspirate; or contraindication to the interruption of anticoagulation [7,8]. According to the American College of Radiology (ACR) Criteria, the appropriateness of CT angiography (CTA) and angiography is comparable in cases of upper GIB with failure to localize the bleeding source by endoscopy [9]. The sensitivity of CTA in these cases approaches 81% in high-risk patients (defined as requiring 500 mL of transfusion to maintain vital signs), which decreases to 50% in low-risk patients with a slower bleeding rate [10]. In cases of lower GIB manifested as severe hematochezia, the American Society of Gastrointestinal Endoscopy (ASGE) guidelines recommend performing colonoscopy in hemodynamically stable patients, but upper GI endoscopy to rule out the upper GI bleeding source before colonoscopy in hemodynamically unstable patients [11]. The diagnostic yield of colonoscopy ranges from 45% to 100% in LGIB and is significantly higher than radiologic evaluation with red blood cell scans and angiography [12,13]. Although both modalities have equal appropriateness ratings by the American College of Radiology (ACR) Criteria, CTA was shown to be more accurate in localizing the bleeding source [14]. CTA can diagnose active bleeding at a rate of more than 0.3–0.5 mL/min, which is equivalent to 1–2 units of blood loss during one day, by demonstrating extravasation of contrast material into the bowel lumen [15,16]. A meta-analysis of nine studies of patients with overt GIB showed CTA to have a pooled sensitivity of 89% and a specificity of 85% in detecting the bleeding source [17]. It has been shown that CTA can localize the bleeding source in the upper or lower gastrointestinal tract after negative or inconclusive endoscopies [18,19]. Accumulating data suggest that CTA may have a diagnostic role in patients presenting with acute overt GIB [20]. However, no clear data are available regarding the clinical and laboratory predictors of positive CTA examinations in patients with GIB. Therefore, we aimed to explore the predictors of positive CTA examinations among patients with GIB.

## 2. Methods

We conducted a cross-sectional, retrospective study. All hospitalized patients who were referred for CTA as part of an acute GIB evaluation during a 10-year period were considered eligible and enrolled in the study. The extracted data included demographic, clinical, and laboratory parameters. The study cohort was divided into two groups. The first group included patients with positive CTA examinations, as demonstrated by extravasation of contrast material into the bowel lumen, and the second group included patients who had negative CTA examinations. Then, we assessed the utility of CTA in identifying the bleeding source and performed a comparative analysis to examine whether there are predictors of a positive CTA. All CTA exams were performed within 1 h from the GIB occurrence. The study protocol conforms to the ethical guidelines of the 1975 declaration of Helsinki and was approved by the institutional human research committee. Written informed consent was waived by the local ethical committee due to the retrospective, non-interventional nature of the study.

### 2.1. CTA Protocol

Our institutional CTA protocol includes 3 phases: (1) non-contrast, (2) arterial phase, and (3) delayed phase. The non-contrast phase is used to differentiate hyperdense material (calcifications and surgical grafts) from suspected bleeding. The arterial phase aims to assess bleeding 15–30 s after intravenous injection of 1 mL of contrast per 1 kg body weight. The delayed phase imaging allows assessment of slow filling structures and venous circulation 60–75 s after contrast injection.

### 2.2. Statistical Analysis

Univariate descriptive statistic was used to compare patients with and without active GIB bleeding. Data were reported as mean ± standard deviation for quantitative continuous variables and frequencies (percentages) for categorical variables. Univariate and multivariate logistic regression were used to estimate the odds ratio (OR) and confidence interval (CI) of baseline factors that predict a positive CTA examination. A threshold for statistical significance was set at a *p* value < 0.05. All analyses were performed by an experienced statistician using the statistical analysis software (SAS 9.4 vs. 9.4 Copyright (c) 2016 by SAS Institute Inc., Cary, NC, USA).

## 3. Results

A total of 154 patients were included in our study. Among them, 25 patients (16.2%) had active GIB on CTA (group A), as compared to 129 patients (83.8%) who did not (group B). The average age in group A was 72.4 ± 18.3, as compared to 70.2 ± 16.1 years in group B. The patients’ gender was not different between the two groups. The rates of proton pump inhibitors (PPIs), non-steroidal anti-inflammatory drugs (NSAIDs), aspirin, and new oral anticoagulants used were similar between the groups. On the other hand, clopidogrel and warfarin use were higher in group A (16% and 24%) as compared to group B (7% and 6.2%), respectively. Table 1 demonstrates demographics and baseline characteristics.

### 3.1. Clinical Presentation and Laboratory Data of the Study Cohort

Patients in group A presented more with hematochezia and hematemesis (72% and 12%), as compared to group B (48.8% and 6.2%), respectively. On the other hand, melena presentation was more common in group B (36.4%) vs. 12% in group A. Notably, the cardiovascular clinical parameters were similar among the two groups. Concerning laboratory tests, hemoglobin levels were lower in group A compared to group B (7.9 ± 2.4 g/dL vs. 10.9 ± 2.6 g/dL), respectively. Similarly, the platelet count was lower in group A (216.6 ± 110.5 × 10^3^/uL), vs. 242.1 ± 120.2 × 10^3^/uL in group B. The INR level was higher in group A compared to group B (2.36 ± 2.9 vs. 1.4 ± 1). Interestingly, there was no difference in the Glasgow–Blatchford score between the two groups (13 ± 2.8 vs. 12 ± 3.6), respectively. The AIMS65 scores were all lower in group A compared to group B. However, the American Society of Anesthesiologists (ASA) score of 1 was higher in group A compared to group B; the ASA score 2 was more prevalent in group B; and no difference was found in the score of 3. Moreover, the mean number of packed red blood cells administered was 4.28 ± 4.7 in group A vs. 3.67 ± 5.1 in group B, *p* = 0.29. The sites of bleeding in patients with positive CTA examinations were the stomach in 8%, the small intestine in 36%, and the large intestine in 56% (Table 2).

### 3.2. Univariate and Multivariate Logistic Regression Analyses of Parameters Associated with Positive Bleeding on CT Angiography

In our univariate analysis, we identified four parameters that were associated with active bleeding on CTA, including congestive heart failure (OR 2.47, 95% CI 1.04–5.86, *p* = 0.04), warfarin use (OR 4.76, 95% CI 1.49–15.21, *p* = 0.008), higher INR (OR 1.33, 1.04–1.69, *p* = 0.02), and low albumin level (OR 0.37, 95% CI 0.17–0.79, *p* = 0.01). There was no effect of new oral anti-coagulants and anti-platelet drugs, including dabigatran (OR 1.38, *p* = 0.75), apixaban (OR 0.87, *p* = 0.89), rivaroxaban (OR 5.25, *p* = 0.25), aspirin (OR 0.94, *p* = 0.89), and clopidogrel (OR 2.65, *p* = 0.13). Notably, the higher ASA scores of 2 and 3 were not predictors of positive CTA examinations, as compared to an ASA score of 1 (OR 0.23, 95% CI 0.07–0.79, *p* = 0.02). Moreover, there was no effect of the Glasgow–Blatchford score (OR 1.09, *p* = 0.19) and the different AIMS65 score values. Table 3 and Table 4 demonstrate the univariate analysis of all the parameters that were assessed in our study. On multivariate logistic regression analysis, only the international normalized ratio (INR) and low albumin levels maintained a positive association with an active GIB on CTA (OR 1.34, 95% CI 1.02–1.76, *p* = 0.03) and (OR 0.3, 95% CI 0.12–0.7, *p* = 0.005), respectively. While higher ASA scores were negatively associated with active GIB on CTA, congestive heart failure lost its association (OR 0.41, 95% CI 0.06–2.68, *p* = 0.3) (Table 5).

## 4. Discussion

Acute GIB represents one of the major gastrointestinal causes of hospital admissions, but fortunately, it stops spontaneously in most cases. The rest pose substantial diagnostic and therapeutic challenges, with mortality rates of 14% and 3.9% in cases of ongoing bleeding for acute upper and lower GIB, respectively [3,4,6]. Although international associations have published guidelines for approaching these life-threatening presentations, little is known about the clinical and laboratory predictors of positive CTA examinations in these patients.

In our study, we identified several interesting findings. Most importantly, in our univariate analysis, ASA scores of 2 and 3 compared to ASA 1 were negatively associated with positive CTA examinations with (OR 0.23, 95% CI 0.07–0.79, *p* = 0.02). Moreover, in our multivariate regression analysis, ASA score 1 compared separately to ASA 2 and 3 retained its significant association with positive CTA examinations with (OR 13.15, 95% CI 2.84–60.96, *p* = 0.001) and (OR 11.1, 95% CI 1.28–96.31, *p* = 0.03), respectively. However, there was no significant difference between ASA 2 and 3 (*p* = 0.86). A thorough search of the English literature did not identify any publications reporting this interesting finding. Probably, patients with ASA score 1 do not seek medical help as early as ASA 2 and over, due to their self-belief that “everything will be okay”, and they present to the hospital with gastrointestinal bleeding only in severe cases that would be identified by the CTA.

Interestingly, only warfarin use was shown on univariate analysis to be associated with a positive CTA examination (OR 4.76, 95%CI 1.49–15.21, *p* = 0.008), but not the new oral anticoagulants (dabigatran, apixaban, and rivaroxaban), clopidogrel, aspirin, and NSAIDs. The association of warfarin was lost on multivariate analysis. Moreover, we could also show that higher INR values were predictive of positive CTA examinations in univariate analysis (OR 1.33, 95% CI 1.04–1.69, *p* = 0.02) and multivariate analysis with (OR 1.34, 95% CI 1.02–1.76, *p* = 0.03).

Systemic anticoagulation with vitamin K antagonists or direct oral anticoagulants and anti-platelets increases the risk of gastrointestinal bleeding and leads to significant morbidity [21]. Bleeding complications occur in warfarin-treated patients with therapeutic and supra-therapeutic INR ranges at the same frequency [22]. Additionally, the recent report of the Safety of Oral Anticoagulants Registry (SOAR) project (a multicenter observational study) reported a median INR of 3.1 (2.2–4.8) in warfarin-treated patients presenting to the emergency department with acute hemorrhage [23]. In patients treated with warfarin and presenting with acute gastrointestinal bleeding, the INR level was shown in multivariate analysis to represent a significant independent predictor of bleeding source identification by endoscopy [24]. Notably, INR values higher than 1.5 represent one of the five variables of the AIMS65 score, predicting hospital mortality in patients with acute gastrointestinal bleeding [25].

Not less importantly, low albumin levels were shown in the univariate and multivariate analyses to be associated with positive CTA examinations with (OR 0.37, 95% CI 0.17–0.79, *p* = 0.01) and (OR 0.3, 95% CI 0.12–0.7, *p* = 0.005), respectively. The importance of low albumin levels (less than 3 gr/dl) for predicting mortality in acute gastrointestinal bleeding patients has led to incorporating this variable in the AIMS65 score [25]. An interesting study by Shafaghi A et al., comparing three different scores, including AIMS65, the Glasgow–Blatchford score, and the Full Rockall score, in predicting adverse outcomes in patients with upper gastrointestinal bleeding, found that albumin, by itself, has higher predictive accuracy than the three risk scores and that adding the albumin variable to the Glasgow–Blatchford score improved its accuracy in predicting adverse outcomes [26]. A study comparing different scores and looking for risk factors for adverse outcomes in acute lower gastrointestinal bleeding found low hemoglobin and low albumin to be the strongest predictors of severe bleeding [27]. In left ventricular assist device (LVAD) patients, low albumin levels are associated with an increased gastrointestinal bleeding risk [28]. A study evaluating the utility of upper endoscopy or capsule endoscopy as the second investigative study in patients with acute lower gastrointestinal bleeding and negative colonoscopy identified several predictors of positive findings in the upper endoscopy group, including serum albumin levels lower than 3 gr/dl. A positive endoscopic finding was reported in 22%, while 16% needed endoscopic or radiologic therapeutic intervention [29]. However, we could not identify studies reporting the importance of low albumin levels in predicting active bleeding in CTA.

Notably, neither the different AIMS65 scores nor the Glasgow–Blatchford score showed a significant correlation with active bleeding in CTA in the univariate analysis. Although the AIMS65 score was described as able to predict in-hospital mortality in people [30], we could not identify a prediction power for active bleeding in CTA. The same holds for the Glasgow–Blatchford score, which can predict the need for treatment of upper-gastrointestinal bleeding [25]; however, it could not predict a positive CTA. A recent study trying to determine predictors for endoscopic intervention and mortality in patients admitted to the intensive care unit with upper gastrointestinal bleeding did not find these scores to be predictive [31].

Hemodynamic instability, manifested as hypotension and tachycardia, had the same prevalence in both groups of patients with and without a positive CTA examination. These findings are contrary to the results of Sun H et al., who reported that the detection rate increased with increased severity of bleeding and hemodynamic instability (25). Although the hemoglobin level was lower in group A compared to group B (7.9 gr% ± 2.4 vs. 10.9 gr% ± 2.6), respectively, it did not reach statistical significance in the univariate analysis (OR 1.95% CI 0.99–1.02, *p* = 0.86).

A risk score by Aoki T et al., did not include hemoglobin in predicting severe lower gastrointestinal bleeding [29]. On the other hand, the hemoglobin level is included in the AIMS65 score as a predictor of mortality [25]. It was also shown by Sengupta and Tapper that hemoglobin could predict 30-day mortality in lower gastrointestinal bleeding [32].

Moreover, congestive heart failure was shown in the univariate analysis to be associated with a positive CTA examination but not in the multivariate analysis; this could be due to either the small sample size of patients with positive CTA examinations or the fact that the significance of congestive heart failure was due to intervening variables, as its significance disappeared when other significant variables were added to the multivariate model.

Our study has two limitations. The first one is its retrospective single-center design. The second limitation, originating from the first one, is that the decision to order CTA was made by the attending physician based on his personal knowledge and experience, an issue that was not uniform among all the cohort patients. This fact could have influenced the rate of a positive CTA. However, the strength of our study is the fact that it is the first study to look for predictors of positive CTA, which may guide practitioners in the proper and correct use of this modality.

In conclusion, this is the first study to look for predictors of positive CTA in patients presenting with gastrointestinal bleeding. The prediction of a positive CTA may be helpful to stratify patients who may benefit from the timely performance of this examination. According to our findings, CTA is justified in patients with ASA score 1, low albumin, and high INR, and proper use of this modality could reduce unneeded and potentially harmful radiation and contrast exposure, save costs, and save resources. Further multicenter prospective studies are warranted to further investigate our findings.

## Figures and Tables

**Table 1 jcm-12-07696-t001:** Demographics and baseline characteristics.

	Group A	Group B
Number of patients	25	129
Age (years), mean ± SD	72.4 ± 18.3	70.2 ± 16.1
Gender, N (%) Male Female	13 (52) 12 (48)	68 (52.7) 61 (47.3)
Weight (kg)	69.9 ± 13.4	77.2 ± 19.1
Body mass index (kg/m^2^)	25.7 ± 3.6	27.7 ± 6.5
Obesity, N (%)	3 (12)	41 (31.8)
Smoking, N (%)	6 (24)	44 (34.1)
Ischemic heart disease, N (%)	12 (48)	42 (32.6)
Status post cerebrovascular accident, N (%)	6 (24)	15 (11.6)
Diabetes mellitus, N (%)	8 (32)	60 (46.5)
Hypertension, N (%)	18 (72)	92 (71.3)
Renal failure, N (%)	0	1 (0.8)
Congestive heart failure, N (%)	13 (52)	39 (30.2)
Cirrhosis, N (%)	0	8 (6.2)
Malignancy, N (%)	8 (32)	23 (17.8)
Medications use, N (%)		
PPI use	11 (44)	56 (43.41)
NSAIDs use	1 (4)	8 (6.2)
Aspirin use	11 (44)	8 (6.2)
Clopidogrel use	4 (16)	9 (7)
Warfarin use	6 (24)	8 (6.2)
Dabigatran use	1 (4)	5 (3.9)
Apixaban use	1 (4)	8 (6.2)
Rivaroxiban use	1 (4)	1 (0.8)
Ticagrelor	0	0
Prasugrel	0	0

**Table 2 jcm-12-07696-t002:** Clinical presentation and laboratory parameters.

	Group A	Group B
Clinical bleeding presentation, N (%) Hematemesis Coffee ground vomiting Melena Hematochezia	3 (12) 1 (4) 3 (12) 18 (72)	8 (6.2) 10 (7.7) 47 (36.4) 63 (48.8)
Normal conscious state, N (%)	25 (100)	118 (91.5)
Syncope, N (%)	3 (12)	20 (15.5)
Blood pressure (mmHg) Systolic Diastolic	113 ± 27.3 62.8 ± 11	116.8 ± 23.3 65.6 ± 13.3
Heart rate (beat per minute)	87.7 ± 15.2	85.2 ± 17.4
Tachycardia, N (%)	4 (16)	24 (18.6)
Hypotension, N (%)	7 (28)	40 (31)
Major bleeding, N (%)	16 (64)	77 (59.7)
Mechanical ventilation, N (%)	4 (16)	20 (15.5)
Packed blood units, mean ± SD	4.3 ± 4.7	3.7 ± 5.1
Hemoglobin (g/dL)	7.9 ± 2.4	10.9 ± 2.6
Leukocytes (×10^e3^/uL)	11.6 ± 6.5	11.4 ± 6.7
Platelets (×10^e3^/uL)	216.6 ± 110.5	242.1 ± 120.2
International normalized ratio (INR)	2.36 ± 2.9	1.4 ± 1
Creatinine (mg/dL)	1.1 ± 0.9	1.5 ± 1.4
Blood urea nitrogen (mg/dL)	29.26 ± 15.54	35.33 ± 26.25
Albumin (g/dL)	2.8 ± 0.7	3.1 ± 0.6
Glasgow–Blatchford score	13 ± 2.8	12 ± 3.6
ASA score categories, N (%) • Score of 1 • Score of 2 • Score of 3	5 (20) 6 (24) 14 (56)	7 (5.4) 70 (54.3) 52 (40.3)
AIMS65 score categories, N (%) • Score of 0 • Score of 1 • Score of 2 • Score of 3 • Score of 4 • Score of 5	2 10 7 5 1 0	24 48 41 12 3 1
Number of packed cells administered, mean ± SD	4.28 ± 4.7	3.67 ± 5.1
Site of active bleeding, N (%) Stomach Small intestine Large intestine	2 (8) 9 (36) 14 (56)	-

**Table 3 jcm-12-07696-t003:** Univariate analysis of parameters associated with active bleeding on CTA.

	Odds Ratio	95% CI	*p* Value
Age	1.01	0.98–1.03	0.59
Male gender	0.97	0.41–2.26	0.94
Weight	0.97	0.95–1	0.08
Body mass index	0.95	0.88–1.02	0.16
Obesity	0.33	0.1–1.1	0.07
Smoking	0.64	0.24–1.68	0.37
Ischemic heart disease	1.91	0.81–4.49	0.14
Status post cerebrovascular accident	2.46	0.86–7.07	0.09
Diabetes mellitus	0.56	0.23–1.37	0.2
Hypertension	1	0.39–2.55	1
Renal failure	1.74	0.02–157.82	0.81
Congestive heart failure	2.47	1.04–5.86	0.04
Cirrhosis	0.28	0.01–5.95	0.41
Malignancy	2.2	0.85–5.66	0.1
PPI use	1.03	0.44–2.42	0.94
NSAIDs use	0.87	0.13–5.74	0.89
Aspirin use	0.94	0.4–2.20	0.89
Clopidogrel use	2.65	0.76–9.28	0.13
Warfarin use	4.76	1.49–15.21	0.008
Dabigatran use	1.38	0.19–10.22	0.75
Apixaban use	0.87	0.13–5.74	0.89
Rivaroxiban use	5.25	0.32–86.78	0.25
Clinical and laboratory presentations			
Bleeding presentation \hematemesis vs. coffee ground Hematochezia vs. coffee ground Melena vs. coffee ground	2.88 2.04 0.52	0.32–26.27 0.32–13.12 0.06–4.19	0.35 0.45 0.54
Altered conscious state	0.2	0.01–4.02	0.29
Syncope	0.83	0.24–2.88	0.77
Blood pressure Systolic Diastolic	0.99 0.98	0.98–1.01 0.95–1.02	0.47 0.34
Heart rate	1.01	0.98–1.03	0.5
Tachycardia	0.9	0.29–2.77	0.86
Hypotension	0.89	0.35–2.28	0.82
Major bleeding	1.2	0.49–2.92	0.82
Mechanical ventilation	1.12	0.36–3.49	0.85
Hemoglobin	1	0.99–1.02	0.86
Leukocytes	1.01	0.95–1.07	0.76
Platelets	0.99	0.99–1	0.35
INR	1.33	1.04–1.69	0.02
Creatinine	0.81	0.53–1.24	0.33
Blood urea nitrogen	0.99	0.97–1.01	0.33
Albumin	0.37	0.17–0.79	0.01

**Table 4 jcm-12-07696-t004:** Univariate analysis of clinical scores associated with active bleeding on CTA.

	Odds Ratio	95% CI	*p* Value
Glasgow–Blatchford score	1.09	0.96–1.24	0.19
ASA score 2 and 3 vs. 1	0.23	0.07–0.79	0.02
AIMS65 score categories 1 vs. 0 2 vs. 0 3 vs. 0 4 vs. 0 5 vs. 0	2.12 1.77 5.09 1.39 5.09	0.48–9.37 0.38–8.26 0.98–26.37 0.04–54.86 0.07–390.11	0.32 0.47 0.06 0.86 0.46

**Table 5 jcm-12-07696-t005:** Multivariate logistic regression analysis.

	Odds Ratio	95% CI	*p* Value
INR	1.34	1.02–1.76	0.03
Albumin	0.3	0.12–0.7	0.005
Congestive heart failure	0.41	0.06–2.68	0.3
ASA score • 1 vs. 2 • 1 vs. 3 • 2 vs. 3	13.15 11.1 0.84	2.84–60.96 1.28–96.31 0.12–5.72	0.001 0.03 0.86

## Data Availability

The data presented in this study are available on request from the corresponding author. The data are not publicly available due to privacy of the patient.

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
