# Peer review of "Clinical Predictors of Gastrointestinal Bleeding Source before Computed Tomography Angiography"

_jcm, 2023, doi:10.3390/jcm12247696_

Round 1

Reviewer 1 Report

Comments and Suggestions for Authors

The paper brings to scientific society an important subject regarding one of the major emergencies in gastroenterology. Is true that many times the endoscopy is negative in patients with active bleeding. The present article describes nicely the characteristics of followed patients and the predictive factors

I would add some remarks about the time between the endoscopy and CT. Different times can alter the results of CT. And also the active treatment used for stopping the bleeding: PPI, Vit K antagonists, plasma, tranexamic acid, etc.

Regarding the references, for nr 8 you can use the newest ESGE guide from 2021.

Author Response

Reviewer 1

The paper brings to scientific society an important subject regarding one of the major emergencies in gastroenterology. Is true that many times the endoscopy is negative in patients with active bleeding. The present article describes nicely the characteristics of followed patients and the predictive factors

I would add some remarks about the time between the endoscopy and CT. Different times can alter the results of CT – Thank you. All CTA exams were performed within 1 hour from the GIB occurrence. The data were added to the method section.

 And also the active treatment used for stopping the bleeding: PPI, Vit K antagonists, plasma, tranexamic acid, etc – Thank you for this comment. Data were added to text, and to table 1.

Regarding the references, for nr 8 you can use the newest ESGE guide from 2021 – Done.

Reviewer 2 Report

Comments and Suggestions for Authors

This manuscript by Maamoun Basheer et al. explored predictors of positive computed tomography angiography (CTA) examination among patients with acute gastrointestinal bleeding (GIB). The authors studied 154 patients with GIB at a single center, and compared patients’ characteristics using logistic regression analysis to explore parameters associated with a positive CTA. They found that high INR and low albumin as strong predictors of active GIB as demonstrated by positive CTA. The authors have done a good job formulating the idea. This is an important question to study and relevant in the management of GIB. The manuscript tries to address knowledge gaps in the field of GIB and provides some insights into risk factors predicting positive CTA in patients with active GIB. The manuscript is well written.

Consider following suggestions for improvement of this manuscript:

-       Discuss about CT angiography protocol in your center and how it was interpreted.

-       Congestive heart failure was shown on univariate analysis to be associated with a positive CTA exam but not on multivariate analysis. Comment on this in the discussion section.

-       What was the timing of CTA compared to timing of GIB.

-       Include the site of bleeding in patients with positive CTA exam.

-       Mention if any of the positive CTA patients had RBC scintigraphy or follow up endoscopy and what was the findings.

Author Response

This manuscript by Maamoun Basheer et al. explored predictors of positive computed tomography angiography (CTA) examination among patients with acute gastrointestinal bleeding (GIB). The authors studied 154 patients with GIB at a single center, and compared patients’ characteristics using logistic regression analysis to explore parameters associated with a positive CTA. They found that high INR and low albumin as strong predictors of active GIB as demonstrated by positive CTA. The authors have done a good job formulating the idea. This is an important question to study and relevant in the management of GIB. The manuscript tries to address knowledge gaps in the field of GIB and provides some insights into risk factors predicting positive CTA in patients with active GIB. The manuscript is well written.

Consider following suggestions for improvement of this manuscript:

-       Discuss about CT angiography protocol in your center and how it was interpreted – Thank you. CTA protocol was added to the manuscript.

-       Congestive heart failure was shown on univariate analysis to be associated with a positive CTA exam but not on multivariate analysis. Comment on this in the discussion section – Thank you. “Moreover, Congestive heart failure was shown on univariate analysis to be associated with a positive CTA exam but not on multivariate analysis, this could be due to either the small sample size of patients with positive CTA exam, or the significance of congestive heart failure was due to intervening variables, as its significance will disappear when other significant variables are added to the multivariate model”, data were added to the discussion section.

-       What was the timing of CTA compared to timing of GIB – Thank you. All CTA exams were performed within 1 hour from the GIB occurrence. Data were added to the text.

-       Include the site of bleeding in patients with positive CTA exam – Thank you. Data were added to the text and table 2.

-       Mention if any of the positive CTA patients had RBC scintigraphy or follow up endoscopy and what was the findings – Thank you. None of the patients had RBC scintigraphy or or follow up endoscopy.

Round 2

Reviewer 2 Report

Comments and Suggestions for Authors

Authors addressed my questions. Manuscript can be published if other reviewers concerns are addressed.